# A Dual Coverage Monitoring of the Bile Acids Profile in the Liver–Gut Axis throughout the Whole Inflammation-Cancer Transformation Progressive: Reveal Hepatocellular Carcinoma Pathogenesis

**DOI:** 10.3390/ijms24054258

**Published:** 2023-02-21

**Authors:** Luwen Xing, Yiwen Zhang, Saiyu Li, Minghui Tong, Kaishun Bi, Qian Zhang, Qing Li

**Affiliations:** Faculty of Pharmacy, Shenyang Pharmaceutical University, 103 Wenhua Road, Shenyang 110016, China

**Keywords:** hepatocellular carcinoma, inflammation-cancer transformation process, bile acids, enterohepatic circulation, BAAT, liver–gut axis

## Abstract

Hepatocellular carcinoma (HCC) is the terminal phase of multiple chronic liver diseases, and evidence supports chronic uncontrollable inflammation being one of the potential mechanisms leading to HCC formation. The dysregulation of bile acid homeostasis in the enterohepatic circulation has become a hot research issue concerning revealing the pathogenesis of the inflammatory-cancerous transformation process. We reproduced the development of HCC through an *N*-nitrosodiethylamine (DEN)-induced rat model in 20 weeks. We achieved the monitoring of the bile acid profile in the plasma, liver, and intestine during the evolution of “hepatitis-cirrhosis-HCC” by using an ultra-performance liquid chromatography-tandem mass spectrometer for absolute quantification of bile acids. We observed differences in the level of primary and secondary bile acids both in plasma, liver, and intestine when compared to controls, particularly a sustained reduction of intestine taurine-conjugated bile acid level. Moreover, we identified chenodeoxycholic acid, lithocholic acid, ursodeoxycholic acid, and glycolithocholic acid in plasma as biomarkers for early diagnosis of HCC. We also identified bile acid-CoA:amino acid *N*-acyltransferase (BAAT) by gene set enrichment analysis, which dominates the final step in the synthesis of conjugated bile acids associated with the inflammatory-cancer transformation process. In conclusion, our study provided comprehensive bile acid metabolic fingerprinting in the liver–gut axis during the inflammation-cancer transformation process, laying the foundation for providing a new perspective for the diagnosis, prevention, and treatment of HCC.

## 1. Introduction

HCC is one of the most serious malignancy tumors threatening human health, the third leading cause of cancer-related death in the world [1,2,3]. Persistent inflammation leading to the formation of the tumor microenvironment is an important factor in the formation of HCC, whose mechanism is very complicated. The morbidity trend of HCC appears to be closely related to hepatitis B (HBV) infection, and it has been reported that HCC patients caused by HBV still account for more than half of the global cases [4,5,6]. Although the inflammation-cancer transformation process of “hepatitis-cirrhosis-HCC” has become a research highlight to reveal the pathogenesis of HCC, there is still no effective clinical treatment strategy. Hence, it has become important to clarify the pathogenesis of the process to achieve early diagnosis of HCC and identify new therapeutic targets.

Among the various endogenous metabolites originating from the co-metabolism of the liver–gut axis, bile acids (BAs) have received increasing attention because of their neoplasm-promoting properties [7,8,9,10]. BAs are synthesized in the liver, and the size and composition of the liver bile acid pool are closely regulated by translocation proteins [11]. When liver organic solute transporter-alpha/beta (OSTα/OSTβ) expression is downregulated, abnormal retention of BAs in hepatocytes within the organism occurs, leading to chronic liver injury [12], while patients diagnosed with HCC between 13 and 52 months concerning bile acid transporter deficiency resulted in a suppression of liver bile acid efflux [13]. In addition to the effect of liver bile acid accumulation on hepatocarcinogenesis, disruption of intestinal bile acid pool homeostasis can contribute to cancer development and a variety of chronic disease phenotypes. Elevated levels of secondary BAs in feces are capable of causing structural and functional abnormalities in the colonic epithelium through various mechanisms, including oxidative damage to DNA, activation of nuclear factor kappa-B, and enhanced cell proliferation [14]. However, depicting the spectrum of BAs and their interactions in plasma, liver, and intestine, covering the entire enterohepatic circulation, during the overall disease course of “health-hepatitis-cirrhosis-HCC” still requires research.

In this paper, based on an *N*,*N*-diethyl-1,4-butanediamine (DEABA) derivatization method for absolute quantification of BAs, the systematic bile acid profiles in plasma, liver, and intestine in the whole progression of HCC have been obtained. Combined with analysis of independent sample *t*-tests, principal component analysis (PCA), orthogonal partial least squares discrimination analysis (OPLS-DA), and bayesian linear discriminant analysis (BLDA), key BAs biomarkers were screened out to distinguish different disease stages, which was valuable for the early diagnosis of HCC. Next, gene set enrichment analysis (GSEA) and the cancer genome atlas (TCGA) database were employed to explore the effect of core genes on the distribution of bile acid pools, which was crucial for promoting the development of HCC. Our study revealed the change of BAs in the liver–gut axis during the inflammation-cancer transformation process and provided a novel perspective for treating HCC.

## 2. Results

### 2.1. Histology Assessment and Total Bile Acid Features in the Inflammation-Cancer Transformation Process

Changes in total bile acid (TBA) levels can reflect the physiological status and injury degree of the organism. Studies have confirmed that the TBA profiles of patients with HCC have unique metabolic characteristics, and the homeostasis of TBA is dependent on liver synthesis and intestinal absorption [15,16]. To elucidate the etiopathogenesis of HCC underlying TBA metabolism disorders, the present study evaluated the various canceration stages of DEN-induced rats based on the results of hematoxylin-eosin (H & E) stained liver tissue sections and quantified the TBA level in rats plasma, liver, and intestine at different stages of HCC progression.

H & E staining showed that hepatocytes began to exhibit severe impairment in the 8th week compared to healthy controls (Figure 1A), termed the hepatitis stage (Figure 1B). The liver tissue was infiltrated with lymphocyte-dominated inflammatory cells, with a small amount of bile duct hyperplasia and localized vascular stasis. The cirrhosis stage occurred in the 12th week (Figure 1C), with an obvious structural disorder of liver lobules, the proliferation of perivenous connective tissue, formation of pseudo lobules with hepatocyte regeneration nodules, and bile duct hyperplasia. The 16th week was the initial stage of HCC (Figure 1D). Microscopically, hepatocyte empty valve degeneration and a small number of adenoid structures were observed, and a large amount of bile duct hyperplasia was visible. At the same time, massive vascular stasis and brownish-yellow pigmentation were observed. The 20th week was described as an advanced HCC stage (Figure 1E). The hepatic tissue showed obvious adenoid structures, all cells had enlarged deep-stained nuclei, and different degrees of vacuolar degeneration were observed.

Based on the histological results, we found that TBA levels significantly increased in all disease groups (Figure 2A). The TBA level of intestinal contents samples gradually decreased with disease progression, which showed an opposite trend to plasma and liver samples (Hepatitis & Cirrhosis vs. Control ** *p* < 0.01; HCC & Advanced HCC vs. Control * *p* < 0.05), while the TBA levels in plasma and liver gradually increased in all stages (* *p* < 0.05, ** *p* < 0.01). Therefore, we speculate that there is a close relationship between the inflammation-cancer transformation process and enterohepatic circulation.

To further analyze the specific reasons for the gradual decrease of TBA levels in the intestine, we subsequently analyzed total primary and secondary BAs in plasma, liver, and intestinal contents. We found that total primary and secondary BAs were markedly elevated in plasma and intestinal contents. However, we observed a specific phenomenon of elevated total primary BAs but decreased secondary BAs in liver samples only (* *p* < 0.05, ** *p* < 0.01). With the development of HCC, total primary BAs in the intestine decline in the advanced HCC stages, in contrast to the continuous increment of total primary BAs in the plasma and liver (Figure 2B). In addition, it is noteworthy that the total secondary BA level in plasma and liver showed an abnormal rebound at the advanced HCC stage, which was not seen in intestinal contents (Figure 2C).

### 2.2. Observing Liver–Gut Axis BAs Environment and Screening HCC Biomarkers for Early Diagnosis

To figure out the key driving BAs for the evolving of HCC, we quantified the changes in the levels of 5 free BAs (cholalic acid, CA; chenodeoxycholic acid, CDCA; ursodeoxycholic acid, UDCA; lithocholic acid, LCA; deoxycholic acid, DCA; Figure 3), and their associated 10 conjugated BAs in plasma, liver, and intestine (Figure 4). The quantitative results of 15 BAs in plasma, liver, and intestinal contents samples from different disease stages of HCC and healthy controls are included in Appendix A, and the results are expressed as mean ± SD.

For free BAs, we found the same trend in three samples, with a significant increase in CA, CDCA, and DCA and a marked decline in LCA (* *p* < 0.05, ** *p* < 0.01). In addition, the different phenomena in UDCA are noteworthy, which were reduced in the liver and intestinal contents but elevated in plasma.

In rodents, free BAs are more likely to be coupled to taurine, the glycine-conjugated BAs accounting for a small proportion of conjugated BAs [17]. The report supports that glycine-conjugated BAs are present at low levels in rats [18]. Due to the low levels and some errors in the quantitative analysis, individual disease groups did not show significant differences compared to the control group. However, from an overall perspective, glycocholic acid (GCA), glycochenodeoxycholic acid (GCDCA), glycodeoxycholic acid (GDCA), glycolithocholic acid (GLCA), and glycoursodeoxycholic acid (GUDCA) all showed similar trends to their prototypes in three samples (Figure 4A–E). Taurocholic acid (TCA), taurochenodeoxycholic acid (TCDCA), tauroursodeoxycholic acid (TDCA), taurolithocholic acid (TLCA), and tauroursodeoxycholic acid (TUDCA) showed consistent trends concerning prototypic BAs only in plasma and liver. Surprisingly, all five taurine-conjugated BAs were reduced in the intestine and found a progressive decrease in TCA, TUDCA, and TDCA with disease progression (* *p* < 0.05, ** *p* < 0.01, Figure 4F–J).

Next, the association between discrepancies in bile acid levels and inflammatory-cancer transformation was established by two multivariate modeling approaches, PCA and OPLS-DA. The results showed that they were distinguished by respective disease stages. As the HCC progresses, the PCA score plot demonstrated a definite trend, confirming the potential of BAs to predict disease staging (Figure 5A–C). Next, combining the contribution degree of OPLS-DA (VIP > 1) and the significance of independent *t*-test (*p* < 0.05), CDCA, LCA, UDCA, and GLCA in plasma, and CDCA in liver and intestine were seen as biomarkers that have a positive role in the early diagnosis of HCC (Figure 5D–F). To date, liver biopsy is currently the gold standard for early diagnosis of HCC, but patient acceptance of this standard invasive technique is poor. A BLDA diagnostic model was constructed by CDCA, LCA, UDCA, and GLCA in plasma to achieve non-invasive detection. The coefficients of the four biomarkers and constants in the BLDA diagnostic model are listed in Table 1. By substituting the bile acid concentrations into the respective equations, the probability of being classified in the corresponding disease group was calculated. The result indicated a reliable model; 86.7% of the samples could be correctly distinguished (Appendix A).

### 2.3. BAAT was Associated with Altered Composition of the Intestinal BA Pool and Disruption of Enterohepatic Circulation

To explore the potential mechanisms of bile acid metabolism changes in HCC patients and to screen out valuable key target genes, 373 HCC samples and 50 healthy samples from the TCGA database were involved in the present analysis. GSEA enrichment analysis was used to screen out 15 gene sets related to the biological functions of bile acid (Appendix A, Appendix A). We obtained 125 genes from 15 gene sets to import into the STRING database to complete the visualization of Protein-Protein Interaction (PPI) Networks with a confidence level > 0.4 (Figure 6). Finally, Cytoscape software was applied to calculate the key node genes based on the cyto Hubba plug-in and maximal clique centrality (MCC) algorithm, the most core gene BAAT was obtained, ranking first.

BAAT is the final modification before catalyzing the generation of conjugated BAs from free BAs into the enterohepatic circulation [19]. Evidence indicates that BAAT promotes glycine-conjugating BAs with extremely low efficiency but efficient conjugating with taurine in rats [20,21]. The TCGA database supports that BAAT is significantly under-expressed in HCC cases (** *p* < 0.01, Appendix A), suggesting that BAAT deficiency is partly responsible for the decrease in taurine-conjugated BAs in the intestine, which would alter the composition of the intestinal bile acid pool and increase its toxicity, thereby promoting the progression of inflammation to HCC.

## 3. Discussion

In recent years, HBV infection has progressively developed into a major cause of HCC. At the same time, 80–90% of new cases occur in the context of cirrhosis, suggesting that hepatitis and cirrhosis play important roles in the precancerous liver environment [22,23]. It is confirmed through research that early diagnosis of HCC by monitoring BAs may improve prognosis and the feasibility of curative treatment [24]. Meanwhile, the bidirectional communication of the liver–gut axis is an essential part of coordinating the dynamic balance of the bile acid pool in the body [25]. However, there are few existing articles describing whole bile acid profiling in enterohepatic circulation during the process of “hepatitis-cirrhosis-HCC”. The pathogenesis of HCC has not been clear till now. We clarified the four disease stages of HCC development based on the previous literature [26,27] and histopathological analysis, first achieving the dual coverage monitoring of the dynamic changes of bile acid levels and distribution during the enterohepatic circulation and the evolution of “hepatitis-cirrhosis-HCC”, and found that the imbalance of the enterohepatic circulation system was the key driver of the inflammation-cancer transformation process, which contributes to cognitive the pathogenesis of HCC.

This study indicated a significant sludge of BAs in the liver–gut axis, while TBA levels in plasma and liver are positively correlated with HCC progression. High levels of bile acid environment have been known to induce reactive oxygen species production and apoptosis in hepatocytes, further leading to impaired liver function [28]. It was accepted that the gradual accumulation of TBA is a major risk factor for the development of HCC, while it is well established that TBA levels and enterohepatic circulation profoundly influence each other [29]. Enterohepatic circulation is the process by which BAs pass from the liver to the intestine and then return to the liver through reabsorption from the portal vein [25,30]. The above process is intricately linked to processes that mainly undergo extensive feedback and feed-forward regulation by specialized absorption and excretion transport systems in the liver and intestine [31]. Furthermore, defective expression and function of bile acid export, as well as reabsorption, have been recognized as important causes of progressive cholestasis in the liver and plasma [32,33]. BAs in the above process are circulated through specialized absorption and excretion transport systems in the liver and intestine. Bile salt export pump (BSEP) and multidrug resistance-associated protein (MRP2) are key transport proteins for the hepatic efflux of BAs, while sodium bile acid/taurocholic synergistic polypeptide (NTCP) and organic anion transport peptide (OATP) are the main transport proteins in the liver responsible for uptake of circulating BAs in the portal vein [34,35]. Reports on patients with HCC also indicate that BSEP, MRP2, NTCP, and OATP expression is downregulated [29,36], corroborating the disruption of enterohepatic circulation in the development of HCC.

The intestine is the site of secondary BA synthesis. Primary BAs synthesized in the liver are further metabolized in the intestine [37]. We provide dysregulation of the primary and secondary BAs in the liver–gut axis, revealing a unique metabolic regulation of BAs in the intestine. The organic solute transporter-alpha/beta (OSTα/OSTβ) are exporters of BAs from the intestine and are an important link in enterohepatic circulation [38]. It has been confirmed in the literature [39] that the absence of OSTα/OSTβ expression causes an increased level of BAs in the intestinal contents as well as in the small intestine. Our quantitative results showed that total secondary BAs were most significantly elevated in the intestine, in addition to being equally elevated in plasma but reduced in the liver, a characteristic phenomenon that likewise suggests a deficiency of the liver bile acid transport system.

Mechanisms underlying the failure of the intestinal barrier and the development of a leaky gut are not fully understood. Still, abnormal retention of toxic BAs is recognized as an important contributing factor [40,41,42]. Secondary BAs are generated from primary BAs through reactions such as 7α-dehydroxylation, so they have the highest hydrophobicity compared to all BAs, a property thought to be linked to hepatotoxicity [43]. On the other side, secondary BAs and their derivatives are a major component of the intestinal bile acid pool, and their elevation represents a change in the toxicity of the intestinal bile acid pool [44]. With the progressive development of HCC, we concluded that due to the large accumulation of secondary BAs in the intestinal epithelium, the intestinal permeability is altered, which eventually causes intestinal fistula. Therefore, we believe that the phenomenon of an abnormal rebound of total secondary bile acids in plasma and the liver is caused by the development of intestinal fistula and the massive efflux of toxic substances accumulated in the intestine at the advanced HCC stage. The above processes also coincided with a progressive decrease of total and secondary BAs in the intestine of the disease group.

CA and CDCA are two primary BAs, and DCA and LCA are secondary BAs from their conversion, respectively. According to the report that the hydrophobic-hydrophilic balance of BAs is closely related to metabolic homeostasis in vivo [45], more hydrophobic BAs can act as cancer promoters and further amplify the development of HCC [46,47]. The high hydrophobicity of CDCA and DCA makes them cytotoxic and pro-inflammatory [48,49]. CA is not highly hydrophobic, but studies have shown that feeding mice with CA increases the size and hydrophobicity of the bile acid pool while causing cholestasis and hepatic steatosis [50]. LCA also has hydrophobic properties, but that’s a small fraction of BAs. UDCA is a primary bile acid in rats, a non-toxic hydrophilic bile acid [51]. Evidence supports the ability of UDCA to accelerate enterohepatic circulation and its cytoprotective properties [52,53]. Therefore, the elevation of CA, CDCA, and DCA in the liver and intestine and the downregulation of LCA and UDCA imply a hydrophobic change in the composition of BAs and a progressive accumulation of toxic BAs that inhibit the enterohepatic circulation. Bile flow is primarily dependent on the drive of conjugated BAs. Congenital defects in BA conjugating can lead to malabsorption of fat-soluble vitamins and, thus, severe liver disease [54,55]. BAAT is the key enzyme capable of mediating bile acid coupling [19]. As mentioned earlier, it has been demonstrated that BAAT -/- mice are almost completely devoid of taurine-conjugated BAs in the liver, suggesting that BAAT is the primary taurine-coupled enzyme in mice [56,57]. Our figures showed that the TBA level in the intestines remained significantly elevated. At the same time, all the taurine-conjugated BAs were continuously reductive in the intestine of model rats. We speculate that the down-regulation of BAAT expression is the key reason for the above phenomenon. Consistent with this, the gene enrichment results confirm our previous speculation about the variation of taurine-conjugated BAs level in the intestine.

## 4. Materials and Methods

### 4.1. Reagents

Acetonitrile, isopropanol, and methanol were purchased from Fisher Scientific (Fair Lawn, NJ, USA), while formic acid, dimethyl sulfoxide, and acetone were purchased from Yuwang Co. Ltd. (Yucheng, China). The distilled water used in the experiments was purchased from Wahaha Group Co., Ltd. (Hangzhou, China). DEN used in animal experiments was purchased from Sigma-Aldrich (St. Louis, MO, USA).

The commercial standards selected for this study, the bile acid used for quantitative analysis, their abbreviations, CAS numbers, and manufacturers are included in Appendix A.

### 4.2. Animals

For this study, Wistar male rats, weighing 100 ± 20 g, purchased from by the Animal Ethical Committee of Changsheng Biotechnology (IACUC No. CSE202106002), were used and provided a constant relative humidity of 65 ± 15% and a temperature of 23 ± 2 °C environment with 12 h-light dark cycles. At the same time, the rats have full access to food and water. The rats were fed and acclimatized to their environment for one week prior to the experiment. Then, 64 rats were randomly divided into two groups, the HCC model group and the healthy control group. Rats in the model group (*n* = 32) were injected intraperitoneally with DEN solution at a dose of 70 mg/kg once a week for 10 weeks, while rats in the control group (*n* = 32) were injected intraperitoneally with an equal volume of saline as a control.

### 4.3. Histopathological Analysis

Liver tissue sections were deparaffinized with xylene and dehydrated in ethanol. Making tissue into 3 µm slice samples and then stained with H&E. Images were acquired using a NIKON digital sight DS-FI2 imaging system after observation with a NIKON Eclipse ci optical microscope.

### 4.4. UFLC-MS/MS Conditions for Quantitation of BAs

A previously published method by our group was used to quantify the BAs [58]. The method was based on a polar response homogeneous dispersion strategy with DEABA labeling, which reduces the polarity and response gap of the analytes and improves selectivity compared to non-derivatization. The ultra-performance liquid chromatography—tandem mass spectrometer (UPLC-MS/MS) systems and chromatographic column were used for the analysis, and liquid phase conditions can be found in previous methods. The positive ion gradient elution program was: 0.01–10.00 min, 20%B→50%B; 10.00–17.00 min, 50%B→85%B; 17.00–22.00 min, 85%B→90%B. The negative ion gradient elution program was 0.01–4.00 min, 20%B→35%B; 4.00–6.00 min, 35%B→70%B; 6.00–10.00 min, 70%B→85%B. 10.00–10.10 min, 85%B→90%B, and continued with 90% B running at 10.10–12.00 min.

We used the electrospray ionization (ESI) source in both positive and negative ion form to accomplish the analysis and determination of BAs by multiple reaction monitoring (MRM) modes. The ion spray voltage was 5500 V(+)/4500 V(−), and the other parameters of the mass spectrum were as follows: curtain gas (N_2_), 20 psi; nebulizer gas (gas 1, N_2_), 50 psi; heater gas (gas 2, N_2_), 50 psi; and source temperature, 500 °C(+)/500 °C(−). The corresponding mass spectrometer (MS) parameters for the 15 BAs can be found in Appendix A.

### 4.5. Sample Collection and Pretreatment

For plasma samples, the whole blood samples were collected from each group following forbidden food for 12 h, placed in heparinized sterile eppendorf tubes, and centrifuged at 10,142× *g* for 10 min at 4 °C to transfer plasma. Then, BAs were extracted from plasma samples as described in the previous method [58].

For liver samples, rats in each group were killed by cervical dislocation after plasma collection. Liver tissue was immediately peeled out, bathed in physiological saline, blotted through filter paper, and transferred to a dry ice box soon afterward. Liver tissue samples (50.00  ±  0.50 mg each) were homogenized in 100 μL physiological saline for 5 cycles (5 s at 300 w, with 3 s between each cycle) by using an ultrasonic cell disruptor (JY92-IIDN, SCIENTZ, Zhengjiang, China) in an ice bath. One liver homogenate was added to 10 µL of internal standard and 10 µL of methanol, the same internal standard used for plasma samples. After vortex shaking for 30 s, 500 µL of precipitated protein reagent, methanol:isopropanol (*v*/*v*, 1:2), was added. The homogenate was centrifuged (4 °C, 10,142× *g*) with vortex shaking for 5 min for 10 min, and the upper layer was dried under a stream of nitrogen. The dried liver samples were derivatized in the same manner as the plasma samples and then subjected to subsequent analysis.

For intestinal contents samples, on the day before the rats were killed, the rats were placed in metabolic cages to collect 24 h intestinal contents. The collected intestinal contents samples were lyophilized for 48 h and ground into powder. 50 mg ±  0.50 mg was taken from intestinal contents lyophilized powder and spiked with 500 µL of physiological saline, then vortexed for 10 min to obtain Intestinal contents homogenate. The pretreatment procedure for intestinal contents samples was approximately the same as for liver samples. The difference is that for protein precipitation, 600 µL of methanol:acetonitrile:acetone (*v/v/v*, 1:1:1) was added to the intestinal contents sample, and the supernatant before drying was filtered through 0.22 μm organic filter membrane. The dried intestinal contents sample were derivatized in the same manner as the plasma samples and then subjected to subsequent analysis.

### 4.6. Gene Enrichment Analysis

We collected samples from The TCGA genomic data commons data portal (https://portal.gdc.cancer.gov/ (accessed on 15 September 2022)) and obtained their RNA sequencing fragments per kilobase million data.

In this study, we selected the gene sets associated with biological functions of bile acid (shown in Appendix A) from the GSEA data set (https://www.gsea-msigdb.org/ (accessed on 5 September 2022)) and performed enrichment analysis between the two groups by GSEA software (version 4.2.3). Among them, gene sets whose *p*-value < 0.05, false discovery rate (FDR) < 0.05, and normalized enrichment score (NES) > 1.5 were collected for subsequence procession. We visualized the PPI network using STRING 11.5 (https://cn.string-db.org/ (accessed on 18 November 2022 )) and the cytoHubba plug-in of Cytoscape (version 3.9.1) software for screening key genes.

### 4.7. Statistical Analysis

The generated raw data files were processed using the Analyst^®^ application (version 1.5.1, AB SCIEX^™^, Foster City, CA, USA), based on which standard curves were created, and all BAs were quantified. The significant differences between the experimental groups were determined using the SPSS Statistics (version 26.0, CHI, Chicago, IL, USA) and GraphPad Prism (version 9.2.0, GraphPad Software Inc., San Diego, CA, USA). The BLDA discriminant analysis was carried out with SPSS software, while PCA and OPLS-DA analysis used the SIMCA-P program (version 14.1, Umetrics, Malmö, Sweden). When the *p*-value < 0.05 or less, we considered the data evidently different and statistically significant.

## 5. Conclusions

In this study, we achieved a dual coverage monitoring of the bile acid profile in the liver–gut axis throughout the whole inflammation-cancer transformation progression. We found that the enterohepatic circulation is disrupted during HCC development after intensively researching the differences in levels of TBA, primary/secondary BAs, and single BAs. Next, we used GSEA gene enrichment analysis to obtain the key node gene BAAT, which dominates the synthesis of taurine-conjugated BAs in rats. We also validated our specific phenomenon of taurine-conjugated BAs in the intestine.

In summary, our results suggest that the disruption of the enterohepatic circulation in the internal environment is an important factor dominating the inflammation-cancer transformation process. The lack of BAAT may be one of the potential mechanisms interrupting the enterohepatic circulation. Additionally, we developed the BLAD diagnostic model, and found that GLCA, CDCA, UDCA, and LCA in plasma samples can be used as biomarkers to distinguish the different disease stages of HCC, enabling early diagnosis of HCC from the perspective of non-invasive detection. However, immunotherapy has been a hot research topic for treating HCC. It has been recently suggested that regulatory T cells, the most abundant immunosuppressive cell population of the HCC-related tumor microenvironment, might suggest a potential target for HCC immunotherapy [59]. Evidence supports that intestinal flora influences the differentiation, accumulation, and function of regulatory T cells [60], and the influence of intestinal flora on BAs metabolism is well established [61,62]. In future studies, it is of great interest and necessity to focus on the link between BAs metabolism, intestinal flora, and the immune cell population of the tumor microenvironment, which will contribute to the further development of HCC therapy.

## Figures and Tables

**Figure 1 ijms-24-04258-f001:**
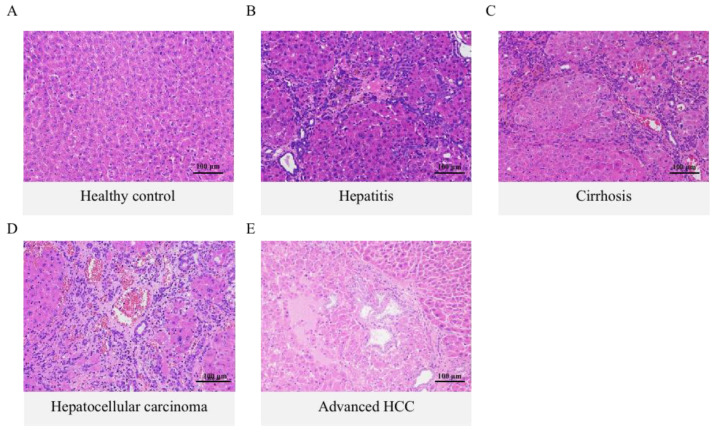
Histological examination of the liver in DEN-induced HCC rats during the inflammation-cancer transformation process. (**A**) The healthy control group showed a normal arrangement of liver cells and a regular structure of liver lobules. (**B**–**E**) Representative micrographs of liver tissue sections at various carcinogenic stages (H & E stain, ×200).

**Figure 2 ijms-24-04258-f002:**
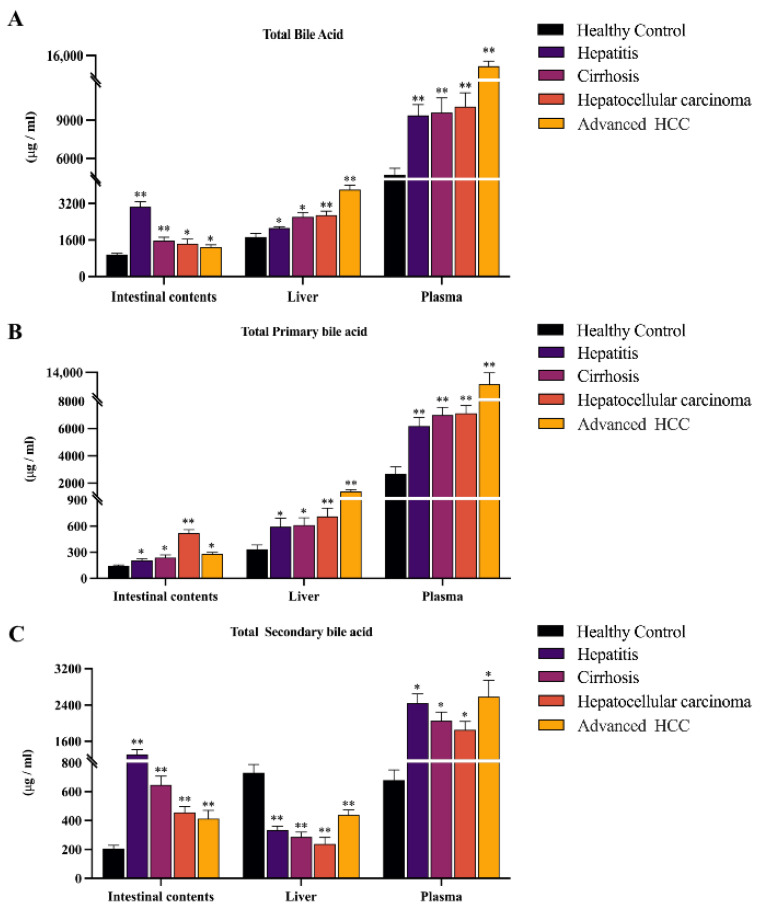
The variations of TBA, total primary BAs, and total secondary BAs in different specimens of DEN-induced HCC rats at healthy, hepatitis, cirrhosis, HCC, and advanced cancer stages. (**A**) TBA concentrations were significantly higher than blank controls in plasma, liver, and intestinal contents samples, and positively correlated with disease progression in plasma and liver. (**B**) The level change of total primary BAs was basically identical between plasma, liver, and intestinal contents, which were significantly elevated at all stages. (**C**) Total secondary BA levels were elevated in plasma and intestinal contents samples while significantly decreased in liver samples over the full course of the disease. The figure indicates statistical significance compared with healthy controls by *t*-tests. * *p* < 0.05, ** *p* < 0.01.

**Figure 3 ijms-24-04258-f003:**
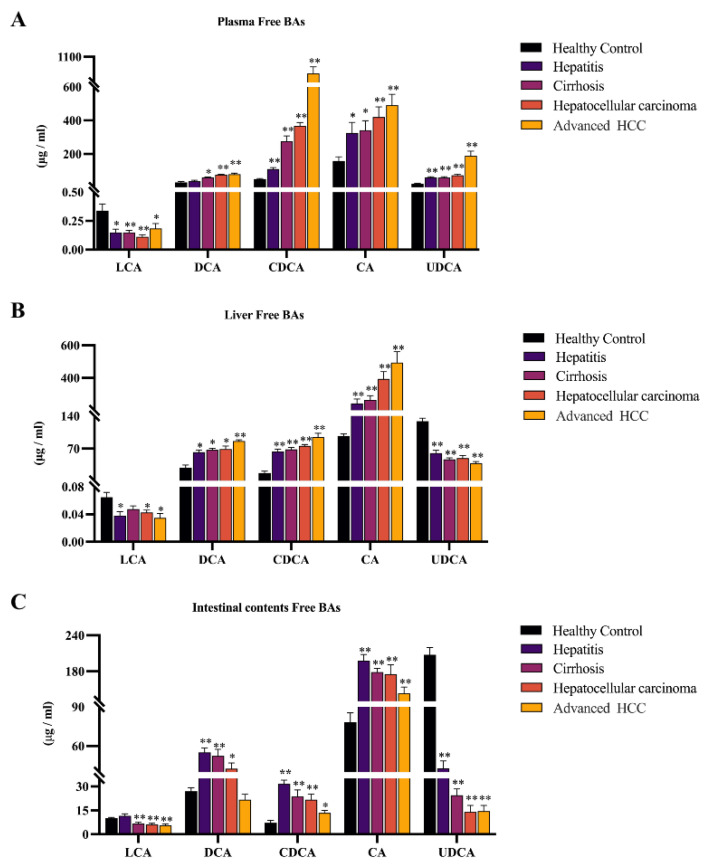
Significant changes in the composition of the free bile acid pool in DEN-induced HCC model rats. Hydrophobic BAs were elevated, and hydrophilic BAs were downregulated in the liver (**B**) and intestinal contents (**C**). In addition, plasma (**A**) was elevated except for LCA. The figure indicates statistical significance compared with healthy controls by *t*-tests, * *p* < 0.05, ** *p* < 0.01.

**Figure 4 ijms-24-04258-f004:**
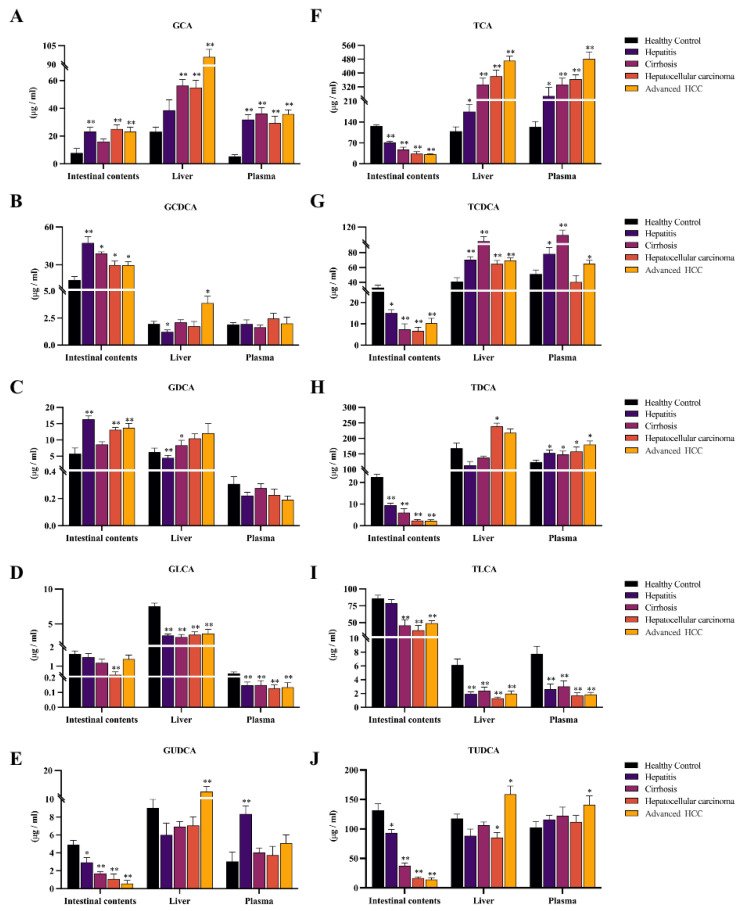
The changes of conjugated BAs in the plasma and liver of DEN-induced model rats were approximately the same as those of prototype BAs. However, the trend in intestinal contents was the opposite, where all taurine-conjugated BAs were significantly reduced. (**A**–**E**): Distribution of glycio-conjugated BAs in three samples. (**F**–**J**): Distribution of taurine-conjugated BAs in three samples. The figure indicates statistical significance compared with healthy controls by *t*-tests, * *p* < 0.05, ** *p* < 0.01.

**Figure 5 ijms-24-04258-f005:**
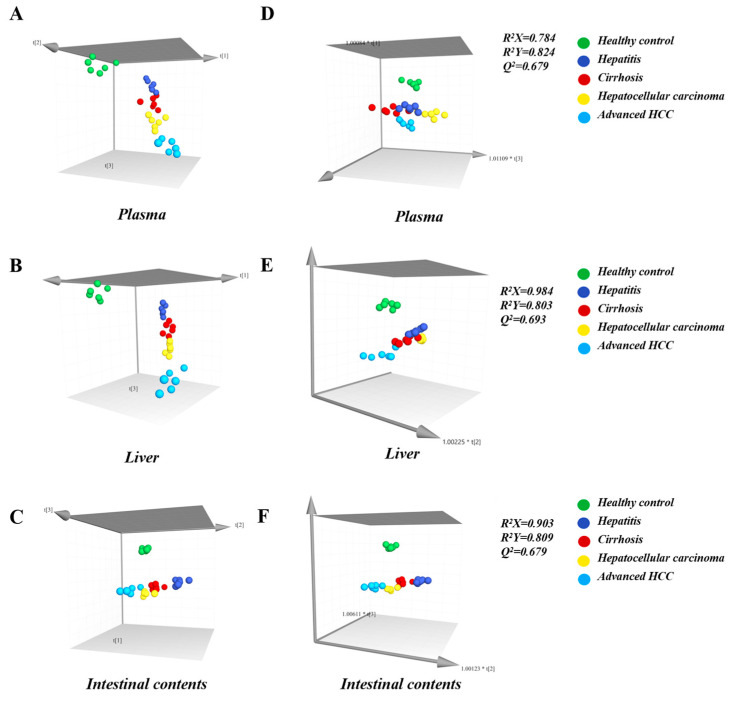
(**A**–**C**): PCA score plots and (**D**–**F**): OPLS-DA score plots of plasma, liver, and intestinal contents samples. In the PCA score plot, the distance between points indicates the difference between samples. The fact that the disease groups are clustered together and separated from the healthy groups indicates that the different groups in the experiment can be well distinguished from each other. R^2^X and R^2^Y in the OPLS-DA score plot indicate the explanation rate of the proposed model for the X and Y matrices, respectively, and Q^2^ marks the predictive power of the model. Usually, the values of R^2^X, R^2^Y, and Q^2^ are higher than 0.5 can indicate that the model fits with acceptable accuracy.

**Figure 6 ijms-24-04258-f006:**
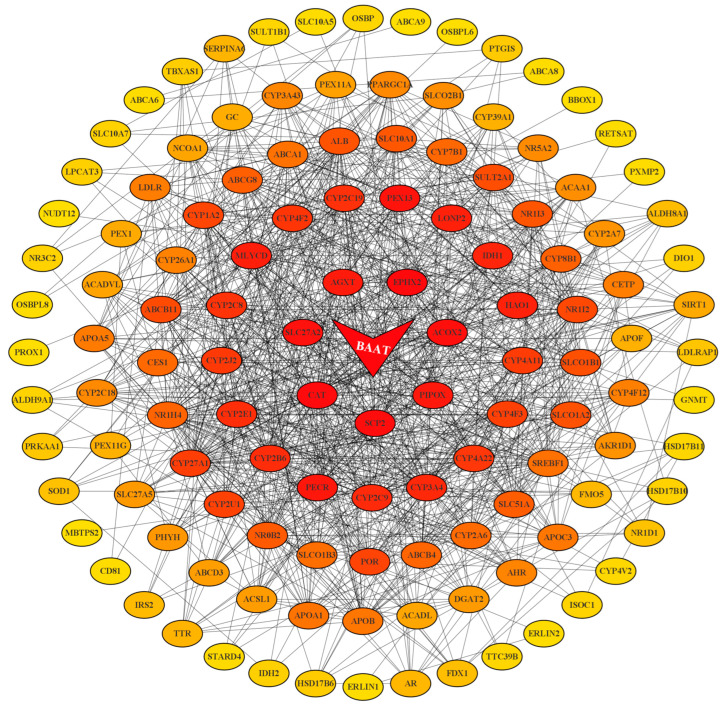
PPI network construction for 125 enriched genes using Cytoscape. The lines in the network diagram represent the connectivity of each gene, and the color represents their scores, with darker nodes representing higher scores and higher rankings. BAAT is the top-ranked most critical node gene based on the MCC algorithm, located in the center of the network diagram.

**Table 1 ijms-24-04258-t001:** The classification function coefficient by BLDA. The larger the constant of BAs, the more significant the effect of abnormal metabolism of this substance on tumorigenesis.

	Health Control	Hepatitis	Cirrhosis	Hepatocellular Carcinoma	Advanced HCC
CDCA	0.006	0.041	0.126	0.169	0.489
LCA	28.368	8.839	16.390	6.180	9.522
UDCA	−0.026	0.058	0.018	0.038	0.097
GLCA	37.047	3.851	−4.336	−5.770	−32.010
Constant	−17.575	−6.589	−20.411	−34.025	−264.774

## Data Availability

All the data used to support the findings of this study are available from the corresponding author upon reasonable request.

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
