# Peer review of "A Dual Coverage Monitoring of the Bile Acids Profile in the Liver–Gut Axis throughout the Whole Inflammation-Cancer Transformation Progressive: Reveal Hepatocellular Carcinoma Pathogenesis"

_ijms, 2023, doi:10.3390/ijms24054258_

Round 1
Reviewer 1 Report
In this study the authors tried to explore the dysregulation of bile acid homeostasis and its potential role in the pathogenesis of the inflammatory-cancerous transformation process. They reproduced the developing of HCC through a N-nitrosodiethyla-mine (DEN)-induced rat model in 20 weeks, and achieved the monitoring of bile acid profile in plasma, liver, and intestine during the evolution of "hepatitis-cirrhosis-HCC" by using UPLC-MS/MS for absolute quantification of bile acids. They observed differences in level of primary and secondary bile acids in plasma, liver and intestine when compared to controls, particularly a sustained reduction of intestine taurine-conjugated bile acid level. Moreover, CDCA, LCA, UDCA and GLCA in plasma were identified as biomarkers for early diagnosis of HCC. They concluded that bile acid metabolic fingerprinting during the inflammation-cancer transformation process, can provide new perspective for the diagnosis, preventing and treating of HCC.
The study is of interest. The authors however, did not provide sufficient characterization and description of immune cell population characterizing tumor microenvironment (TME). This is currently a very relevant feature as it might suggest potential targets of new emerging immunotherapies for HCC. In particular, it has been recently suggested that regulatory T cells (Treg) which are the most abundant immunosuppressive cell population of the HCC-related TME, might explain the different risks of HCC according to the different cirrhosis etiology, as recently described (Hepatocellular carcinoma in viral and autoimmune liver diseases: Role of CD4+ CD25+ Foxp3+ regulatory T cells in the immune microenvironment. World J Gastroenterol. 2021 Jun 14;27(22):2994-3009. ). The authors should discuss how their findings and bile acid data are linked to the important role of immune cell in the TME.
Author Response
Thank you very much for your valuable questions and suggestions. The manuscript has been carefully revised according to your advice. Here are our responses to your comment.
Responses to Reviewer 1:
Comments 1: The authors however, did not provide sufficient characterization and description of immune cell population characterizing tumor microenvironment (TME). This is currently a very relevant feature as it might suggest potential targets of new emerging immunotherapies for HCC. In particular, it has been recently suggested that regulatory T cells (Treg) which are the most abundant immunosuppressive cell population of the HCC-related TME, might explain the different risks of HCC according to the different cirrhosis etiology, as recently described (Hepatocellular carcinoma in viral and autoimmune liver diseases: Role of CD4+ CD25+ Foxp3+ regulatory T cells in the immune microenvironment. World J Gastroenterol. 2021 Jun 14;27(22):2994-3009. ). The authors should discuss how their findings and bile acid data are linked to the important role of immune cell in the TME.
Response: Thank you for this valuable comment, as you suggested, we added “However, immunotherapy has been a hot research topic for the treatment of HCC. It has been recently suggested that regulatory T cells (Treg), which are the most abundant immunosuppressive cell population of the HCC-related tumor microenvironment (TME), might suggest a potential target for HCC immunotherapy [60]. Evidence supports the intestinal flora influences the differentiation, accumulation and function of Treg cells [61] and the influence of intestinal flora on BAs metabolism is well established [62,63]. In future studies, it is of great interest and necessity to focus on the link between BAs metabolism, intestinal flora and the immune cell population of the TME, which will contribute to the further development of HCC therapy.” in the section “Conclusion” (Pg13 line 476-485) to present our outlook for future research and cited the reference you recommended to us. We have been looking for a link between BAs and the immune microenvironment since we received your comments, and we found that intestinal flora is capable of linking BAs and Treg. However, as you mentioned, immunotherapy is revolutionizing the treatment of cancer, and this is an area that deserves to be studied in depth. In the future, we will carefully investigate the close connection between BAs and the important role of immune cell in the TME. Thank you very much for your new perspective.
In order to facilitate your review, we have included the added References below:
- Granito A.; Muratori, L.; Lalanne, C.; et al. Hepatocellular carcinoma in viral and autoimmune liver diseases: Role of CD4+ CD25+ Foxp3+ regulatory T cells in the immune microenvironment. World J Gastroenterol.2021, 27, 2994-3009, doi:10.3748/wjg.v27.i22.2994.
- Tanoue, T.; Atarashi, K.; Honda, K. Development and maintenance of intestinal regulatory T cells. Nat Rev Immunol.2016, 16, 295-309, doi:10.1038/nri.2016.36.
- Jia, W.; Xie, G.; Jia, W. Bile acid-microbiota crosstalk in gastrointestinal inflammation and carcinogenesis. Nat Rev Gastroenterol Hepatol. 2018,15, 111-128. doi:10.1038/nrgastro.2017.119.
- Cai, J.; Sun, L.; Gonzalez, FJ. Gut microbiota-derived bile acids in intestinal immunity, inflammation, and tumorigenesis. Cell Host Microbe. 2022,30, 289-300, doi:10.1016/j.chom.2022.02.004.
We have tried our best to improve the manuscript and made some changes in the manuscript. We appreciate for editors’ and reviewers’ warm work earnestly and hope that the correction will meet with approval. Once again, thank you for all your time involved and for this great opportunity for us to improve the manuscript. We hope you will find this revised version satisfactory.

Reviewer 2 Report
The article “A dual coverage monitoring of the bile acids profile in the liver-gut axis throughout the whole inflammation-cancer transformation progressive: reveal hepatocellular carcinoma pathogenesis” by Xing et al. is an engaging article on a relevant topic, i.e., hepatocellular carcinoma (HCC) as an end-stage chronic liver disease. This research paper attempts to investigate the mechanisms of chronic inflammation that trigger HCC to provide a better understanding of the inflammatory processes, disease activity, and structural and functional impairments in HCC.
The work constitutes a starting point for future research that will allow more accurate and appropriate investigations of medical approaches in this fulminant pathology.
The results are presented and discussed, but with insufficient introspection and objectivity. Figures and tables are not explained in detail and are not well positioned within the article. The conclusion should be formulated more honestly.
The article has several shortcomings as follows:
All references should be double checked.
There is no list of abbreviations to be filled out and carefully reviewed.
Several English and style issues could be improved by a native English speaker in a final reading on the IJMS platform.
Author Response
Thank you very much for your valuable questions and suggestions. The manuscript has been carefully revised according to your advice.
The point-to-point response to the reviewer’s comments are listed as follows:
Responses to Reviewer 2:
Comments 1: The results are presented and discussed, but with insufficient introspection and objectivity. The conclusion should be formulated more honestly.
Response: Thank you for your reminding. As you have suggested, we have rethought and carefully adjusted the “Results” sections. We added “Reports support that glycine-conjugated BAs are present at low levels in rats [18]. Due to the low levels and some errors in the quantitative analysis, individual disease groups did not show significant differences compared to the control group. However, from an overall perspective, glycocholic acid (GCA), glycochenodeoxycholic acid (GCDCA), glycodeoxycholic acid (GDCA), glycolithocholic acid (GLCA) and glycoursodeoxycholic acid (GUDCA) all showed similar trends to their prototypes in three samples (Figure 4A-E). ” in the section “2.2. Observing liver-gut axis BAs environment and screening HCC biomarkers for early diagnosis” (Pg6 line 233-239) to present our reflections on the above phenomenon and to support our speculations and conclusions by citing literatures. At the same time, we have also added a more detailed description of the experimental phenomena. We added “With the development of HCC, total primary BAs in the intestine decline in the advanced HCC stages, in contrast to the continuous increment of total primary BAs in the plasma and liver (Figure 2B). ” in the section “2.1. Histology assessment and total bile acid features in the inflammation-cancer transformation process” (Pg5 line 172-174) to further improve the objectivity and rigor without affecting the conclusions.
We added “The above process is intricately linked processes that mainly undergo extensive feedback and feed-forward regulation by specialized absorption and excretion transport systems in the liver and intestine [32]. Furthermore, defective expression and fuction of bile acid export as well as reabsorption have been recognized as important causes for progressive cholestasis in liver and plasma [33,34]. ”(Pg11 line 393-398) and “Our quantitative results showed that total secondary BAs were most significantly elevated in the intestine, in addition to be equally elevated in plasma but reduced in the liver, a characteristic phenomenon that likewise suggests a deficiency of liver bile acid transport system. ”(Pg12 line 414-417) in the section “Discussion” to better elucidate the functional defects of the transporter system as the main contributor to the abnormal sludge of BAs and propose a link between the intestinal and hepatic transport systems, while supported our speculation through the References 32-34.
Next,we add “Mechanisms underlying the failure of the intestinal barrier and development of a leaky gut are not fully understood, but abnormal retention of toxic bile acids is recognized as an important contributing factor [42,43,44]. Secondary BAs are generated from primary BAs through reactions such as 7α-dehydroxylation, so they have the highest hydrophobicity compared to all BAs, a property that is thought to be linked to hepatotoxicity [45]. On the other side, secondary BAs and their derivatives are a major component of the intestinal bile acid pool, and their elevation represents a change in the toxicity of the intestinal bile acid pool [46]. ” in the section “Discussion” (Pg12 line 418-425) to improve the rationality of our speculation about the phenomenon of intestinal fistula at advanced HCC stage.
In addition, we have presented the “Conclusion” section separately to clearly point out the shortcomings as well as the prospects for future research. Thank you for pointing this out. For our consideration, we hope to get your agreement.
In order to facilitate your review, we have included the added Reference below:
18.Alnouti, Y.; Csanaky, IL.; Klaassen, CD. Quantitative-profiling of bile acids and their conjugates in mouse liver, bile, plasma, and urine using LC-MS/MS. J Chromatogr B Analyt Technol Biomed Life Sci. 2008, 873, 209-217, doi:10.1016/j.jchromb.2008.08.018.
32.Kullak-Ublick, GA.; Stieger, B.; Meier, PJ. Enterohepatic bile salt transporters in normal physiology and liver disease. Gastroenterology. 2004, 126, 322-342, doi:10.1053/j.gastro.2003.06.005.
33.Chen, Y.; Song, X.; Valanejad, L.; et al. Bile salt export pump is dysregulated with altered farnesoid X receptor isoform expression in patients with hepatocellular carcinoma. Hepatology. 2013, 57, 1530-1541, doi:10.1002/hep.26187.
34.Zhang, Y.; Li, F.; Patterson, AD.; et al. Abcb11 deficiency induces cholestasis coupled to impaired β-fatty acid oxidation in mice. J Biol Chem. 2012, 287, 24784-24794, doi:10.1074/jbc.M111.329318.
42.Rodríguez-Nogales, A.; Algieri, F.; Vezza, T.; et al. Calcium Pyruvate Exerts Beneficial Effects in an Experimental Model of Irritable Bowel Disease Induced by DCA in Rats. Nutrients. 2019, 11, 140, doi:10.3390/nu11010140.
43.Fukui, H. Role of Gut Dysbiosis in Liver Diseases: What Have We Learned So Far?. Diseases. 2019, 7, 58. Published 2019 Nov 12, doi:10.3390/diseases7040058.
44.Yu, LX.; Schwabe, RF. The gut microbiome and liver cancer: mechanisms and clinical translation. Nat Rev Gastroenterol Hepatol. 2017, 14, 527-539, doi:10.1038/nrgastro.2017.72.
45.Funabashi, M.; Grove, TL.; Wang, M.; et al. A metabolic pathway for bile acid dehydroxylation by the gut microbiome. Nature. 2020, 582, 566-570, doi:10.1038/s41586-020-2396-4.
46.Ridlon, JM.; Kang, DJ.; Hylemon, PB. Bile salt biotransformations by human intestinal bacteria. J Lipid Res. 2006, 47, 241-259, doi:10.1194/jlr.R500013-JLR200.
Comments 2: The figures and tables are not explained in detail and are not well positioned within the article.
Response: We appreciate your comments and apologize for the lack of detail and inappropriate placement of the figures in the manuscript. We combined Figure 2 and Figure 3 from the original manuscript into a new Figure 2 (Pg4 line 120) to facilitate being able to observe more clearly the information about the changes in total bile acids. Next, we have revised the position of Figures 2B (Pg5 line 174), Figures 2C (Pg5 line 176), Figures 3 (Pg5 line 182), Figures 4 (Pg5 line 183), Figures 5A-C (Pg8 line 256) and Figures 5D-F (Pg8 line 259) in the manuscript in order to make the content of the figures clear to the reader. Meanwhile, we have added group information to Figure 5 and reworked its description as “In the PCA score plot, the distance between points indicates the difference between samples. The fact that the disease groups are clustered together and separated from the healthy groups indicates that the different groups in the experiment can be well distinguished from each other. R2X and R2Y in the OPLS-DA score plot indicate the explanation rate of the proposed model for the X and Y matrices, respectively, and Q2 marks the predictive power of the model. Usually, the values of R2X、R2Y and Q2 are higher than 0.5 can indicate that the model fits with acceptable accuracy.” to add details of the information presented in the PCA score plot and the predictive power of the OPLS-DA model (Pg9 line 309-314) . In addition, we have also revised the description of Figure 6 as “ The lines in the network diagram represent the connectivity of each gene, and the color represents their scores, with darker nodes representing higher scores and higher rankings. BAAT is the top ranked most critical node gene based on the MCC algorithm, which is located in the center of the network diagram.” to give a clearer picture of our experimental results (Pg10 line 354-357). We also modified the description of Table 1, and we added “The larger the constant of BAs, the more significant the effect of abnormal metabolism of this substance on tumorigenesis.” to clarify the information in the table (Pg8 line 269-271). For our consideration, we hope to get your agreement.
Comments 3: All references should be double checked.
Response: Your comments are greatly appreciated and we have rechecked and revised the literature cited in the manuscript carefully. We added Reference 18 (Pg6 line 233) in the “2.2. Observing liver-gut axis BAs environment and screening HCC biomarkers for early diagnosis” section to provide background support for the experimental phenomenon and Reference 32 (Pg11 line 393-396) in the “Discussion” section to complement the description of the transport system of the enterohepatic circulation process. In addition, we added 4 papers (Reference 43-46,Pg12 line 418-425) in the “Discussion” section to rethink the association between secondary BAs and intestinal barrier failure and improve the plausibility and objectivity of the speculations. We added References 60-63 (Pg13 line 476-485) in the “Conclusion” section, which refers to the same model of inflammation-cancer transformation as ours, so we present an outlook for future studies. For our consideration, we would like to obtain your consent.
In order to facilitate your review, we have included the added References below:
18.Alnouti, Y.; Csanaky, IL.; Klaassen, CD. Quantitative-profiling of bile acids and their conjugates in mouse liver, bile, plasma, and urine using LC-MS/MS. J Chromatogr B Analyt Technol Biomed Life Sci. 2008, 873, 209-217, doi:10.1016/j.jchromb.2008.08.018.
32.Kullak-Ublick, GA.; Stieger, B.; Meier, PJ. Enterohepatic bile salt transporters in normal physiology and liver disease. Gastroenterology. 2004, 126, 322-342, doi:10.1053/j.gastro.2003.06.005.
43.Fukui, H. Role of Gut Dysbiosis in Liver Diseases: What Have We Learned So Far?. Diseases. 2019, 7, 58. Published 2019 Nov 12, doi:10.3390/diseases7040058.
44.Yu, LX.; Schwabe, RF. The gut microbiome and liver cancer: mechanisms and clinical translation. Nat Rev Gastroenterol Hepatol. 2017, 14, 527-539, doi:10.1038/nrgastro.2017.72.
45.Funabashi, M.; Grove, TL.; Wang, M.; et al. A metabolic pathway for bile acid dehydroxylation by the gut microbiome. Nature. 2020, 582, 566-570, doi:10.1038/s41586-020-2396-4.
46.Ridlon, JM.; Kang, DJ.; Hylemon, PB. Bile salt biotransformations by human intestinal bacteria. J Lipid Res. 2006, 47, 241-259, doi:10.1194/jlr.R500013-JLR200.
60.Granito A.; Muratori, L.; Lalanne, C.; et al. Hepatocellular carcinoma in viral and autoimmune liver diseases: Role of CD4+ CD25+ Foxp3+ regulatory T cells in the immune microenvironment. World J Gastroenterol. 2021, 27, 2994-3009, doi:10.3748/wjg.v27.i22.2994.
61.Tanoue, T.; Atarashi, K.; Honda, K. Development and maintenance of intestinal regulatory T cells. Nat Rev Immunol. 2016, 16, 295-309, doi:10.1038/nri.2016.36.
62.Jia, W.; Xie, G.; Jia, W. Bile acid-microbiota crosstalk in gastrointestinal inflammation and carcinogenesis. Nat Rev Gastroenterol Hepatol. 2018, 15, 111-128. doi:10.1038/nrgastro.2017.119.
63.Cai, J.; Sun, L.; Gonzalez, FJ. Gut microbiota-derived bile acids in intestinal immunity, inflammation, and tumorigenesis. Cell Host Microbe. 2022, 30, 289-300, doi:10.1016/j.chom.2022.02.004.
Comments 4: There is no list of abbreviations to be filled out and carefully reviewed.
Response: Thank you very much for your valuable suggestions, we have added a list of abbreviations in the section “Abbreviations” (Pg17 line 598-614) and have also carefully reviewed and revised abbreviations that were not used properly to avoid situations where readers would not understand the full text due to unfamiliarity with the abbreviations. We defined abbreviations when they first appeared in the manuscript and ensured that the defined abbreviations were significant and consistent in my use of them. We apologize for any problems with the abbreviations. Thank you for this comment and hope to get your understanding.
Comments 5: Several English and style issues could be improved by a native English speaker in a final reading on the IJMS platform.
Response: Thank you for pointing this out. We apologize for the language problems in the original manuscript. We have polished our manuscript carefully and corrected the grammatical, styling, and typos found in our manuscript. In addition, we improved our language expression problems by reducing the number of inappropriately long sentences in the manuscript and increasing the logic of the language. For our consideration, we hope to get your agreement.
We have tried our best to improve the manuscript and made some changes in the manuscript. We appreciate for editors’ and reviewers’ warm work earnestly and hope that the correction will meet with approval. Once again, thank you for all your time involved and for this great opportunity for us to improve the manuscript. We hope you will find this revised version satisfactory.
